Loss of Farnesoid X receptor (FXR) accelerates dysregulated glucose and renal injury in db/db mice

Qiu Yuxiang 1 2
Kang Ningsu 1
Wang Xi 1
Yao Yao 3
Cui Jun 1
Zhang Xiaoyan xyzhang0916@126.com 4
Zheng Lu ntzl9@163.com 1 2
1 Department of Nephropathy, Nantong Third People’s Hospital , Nantong , China
2 Department of Nephropathy, Affiliated Nantong Hospital 3 of Nantong University , Nantong , China
3 Department of Nephropathy, Affiliated Hospitaland Medical School of Nantong University , Nantong , China
4 Health Science Center, East China Normal University , Shanghai , China
Dong Peixin
Electronic publication date: 2023 Sep 29
Publication date: 2023
Volume: 11
Electronic Location ID: e16155
Received 2023 Jul 13; Accepted 2023 Aug 31
Copyright: ©2023 Qiu et al.
Copyright year: 2023
Copyright holder: Qiu et al.
License: This is an open access article distributed under the terms of the Creative Commons Attribution License, which permits unrestricted use, distribution, reproduction and adaptation in any medium and for any purpose provided that it is properly attributed. For attribution, the original author(s), title, publication source (PeerJ) and either DOI or URL of the article must be cited.
License URL: https://creativecommons.org/licenses/by/4.0/

Keywords: Farnesoid X receptor, Diabetic nephropathy, TGFβ1, Fibrosis, Renal injury

Funding: The authors received no funding for this work.

==============================
Background

End-stage renal disease is primarily caused by diabetic kidney disease (DKD). The Farnesoid X receptor (FXR), a member of the nuclear receptor superfamily, has anti-inflammatory, lipid-lowering and hypoglycemic properties. It also inhibits renal fibrosis. Although its physiological role is not fully understood, it also plays a role in the control of diabetic nephropathy (DN).

Methods

In the present study, we examined male FXR & leptin receptor double knockout mice, in which weight, blood glucose, body fat, and other indicators were monitored. After 6 months of rearing, blood and urine samples were collected and biochemical parameters were measured. Fibrosis was assessed by Masson’s stain, while the assessment of the resuscitation case’s condition was performed using succinate dehydrogenase (SDHA) stain immunohistochemistry, which measures aerobic respiration. Expression of molecules such as connective tissue growth factor (CTGF), SMAD family members 3 (Smad3) and 7 (Smad7), and small heterodimer partner were detected by RT-PCR and Western blotting as part of the application.

Results

FXR knockout decreased body weight and body fat in db/db mice, but increased blood glucose, urine output, and renal fibrosis. Primary mesangial cells (P-MCs) from FXR+/ + mice stimulated with transforming growth factor β1 (TGFβ1) showed significantly higher levels of related fibrosis factors, TGFβ1 and Smad3 mRNA and protein, and significantly reduced levels of Smad7. These effects were reversed by the action of FXR agonist chenodeoxycholic acid (CDCA). P-MCs from FXR−/ − mice stimulated with TGFβ1 resulted in an increase in the expression and protein levels of collagen I and TGFβ1, and the addition of CDCA had no significant effect on TGFβ1 stimulation. However, compared with FXR+/ +db/db mice, the rate of oxygen consumption, the rate of carbon dioxide production, and the rate of energy conversion were increased in FXR−/ −db/db mice, whereas the SDHA succinate dehydrogenase, a marker enzyme for aerobic respiration, was significantly decreased.

Conclusions

These results provide evidence that FXR plays a critical role in the regulation of mesangial cells in DN. The likely mechanism is that aberrant FXR expression activates TGFβ1, which induces extracellular matrix accumulation through the classical Smad signaling pathway, leading to mitochondrial dysfunction.

Introduction

In the ever-shifting landscape of global health, diabetes stands out as a menacing colossus. Among the many tendrils of complications sprouting from diabetes, diabetic kidney disease (DKD) stands out as particularly menacing (Yan et al., 2021). With a staggering 40% to 50% of diabetic individuals falling prey to DKD, its reputation as the principal perpetrator behind end-stage renal disease becomes self-evident (Koay et al., 2021; Umanath & Lewis, 2018). Contemporary research delving into the murky waters of diabetic nephropathy (DN) pathogenesis has shone light upon the critical role of the renin-angiotensin system and glycosylation end products. The current pharmacological armory, boasting of renin-angiotensin system inhibitors like ACE, offers a modicum of resistance against the advancing tide of DN. Yet, their efficacy remains circumscribed, amplifying the clarion call for novel and potent therapeutic interventions.

The sinister tapestry of DKD is embroidered with the threads of irreversible glomerular fibrosis. From the nascent stages, marked by a subtle thickening of the glomerular basement membrane, to the advanced phases characterized by glomerular sclerosis or tubular atrophy, the level of fibrosis remains a reliable barometer of DKD’s relentless progression (Zhang et al., 2022). Although a number of large clinical trials have confirmed that glycemic control can ameliorate diabetic kidney injury, simple glycemic intervention does not effectively prevent the progression of glomerular fibrosis. Glomerular fibrosis is primarily mediated by fibroblasts, has a low physiological content, and is in a quiescent state (Livingston et al., 2023). The primary function of mesangial cells is to provide structural support to the glomerular capillary network while at the same time resisting the high pressure of glomerular fluid caused by ultrafiltration. During DN, mesangial cells that become activated or proliferated will differentiate into myofibroblasts and produce an excessive amounts of extracellular matrix (ECM) (such as type I collagen fibers and fibronectin), which causes thickening of the glomerular basement membrane and obstruction of glomerular capillaries, leading to glomerular sclerosis, which ultimately causes DN (Garcia-Fernandez et al., 2020). There is increasing evidence that mesangial cells play an important role in the pathogenesis of DN (Fu et al., 2019; Tung et al., 2018). Elevated levels of glucose, angiotensin, Amadori products, and advanced glycation end products all affect mesangial cell function to a greater or lesser extent in patients with DN and induce their pathological responses (Nicolas et al., 2019). Studies have shown that abnormal activation of transforming growth factor beta (TGFβ) signaling pathways is the main mechanism of renal fibrosis in DKD. TGFβ1 is a cytokine that has been shown to have multiple biological activities. It is a recognized fibrogenic factor that is an important regulator of ECM deposition leading to glomerular sclerosis (Morita et al., 2022). Smad protein is a regulatory substrate of TGFβ1, Smad protein phosphorylation and nuclear translocation are signs of Smad signaling pathway activation (Kamato et al., 2020). TGF β1 is secreted by mesangial and epithelial cells and has been shown to promote cell division, proliferation, and accumulation of ECM, ultimately leading to renal fibrosis (Peng et al., 2022b).

Mounting evidence underscores the significant interplay between mitochondrial dysfunction and the progression of DKD within innate cells, including the pivotal mesangial cells (Qin et al., 2019; Wei & Szeto, 2019). Mitochondria, often referred to as the powerhouse of eukaryotic cells, are critical hubs of oxidative metabolism. Here, the final oxidation of amino acids, carbohydrates, and lipids takes place, ultimately culminating in the generation of energy. Central to this energy-production mechanism is the synthesis of adenosine triphosphate (ATP) through the tricarboxylic acid cycle and oxidative phosphorylation. These processes, corresponding to the sequential stages of aerobic respiration, harness the potential of key molecules like reduced nicotinamide adenine dinucleotide (NADH) and reduced flavin adenine dinucleotide acid (FADH2) to energize cells (Haddad & Mohiuddin, 2023). The journey of energy production intricately weaves through glycolysis in the cell’s cytosol and the tricarboxylic acid cycle within the mitochondria’s matrix. However, the balance of this powerhouse can be precariously tipped. Factors such as aberrant oxidative stress and a spectrum of inflammatory responses may destabilize mitochondrial homeostasis. The repercussions of such an imbalance are profound: diminished ATP production, potential disruptions in mitochondrial membrane integrity, calcium overloading, and an escalated production of reactive oxygen species (ROS) (Peng et al., 2022a). Excessive ROS is not just a silent tormentor, inflicting direct structural and functional cellular damage; it also manipulates cellular dynamics, influencing proliferation, differentiation, and apoptosis. Its role extends to mediating fibrosis in various organs, as indicated by extensive studies (Nakano et al., 2021; Pan et al., 2021; Yuan et al., 2022). Yet, the intricate tapestry of mitochondrial dysfunction in DKD remains to be completely unraveled, beckoning deeper and more focused investigations.

The farnesoid × receptor (FXR) serves as a pivotal nuclear bile acid receptor, orchestrating the transcriptional regulation of bile acid equilibrium (Kim et al., 2023). More than just overseeing bile acid synthesis and transport in the liver and small intestine, FXR plays a critical role in glucose and lipid metabolism. Its influence is not restricted to just the liver and intestine but extends to organs like the heart, ovary, and notably, the kidney (Xiang et al., 2021; Yan et al., 2021). While its involvement in mitigating DN, modulating blood glucose, and fine-tuning lipid levels is recognized (Tian et al., 2022), the specifics remain elusive. Mesangium cells in the kidney, central to renal fibrosis, exhibit FXR presence, hinting at its potential role in DKD via the TGFβ pathway. FXR also emerges as a key player in energy metabolism, with dysregulation manifesting as mitochondrial dysmetabolism. Interestingly, the FXR agonist, chenodeoxycholic acid (CDCA), shows promise in countering ROS-triggered traumas, and reviving mitochondrial respiratory processes (Gai et al., 2016), though its broader effects on ECM, Smad3, and Smad7 still warrant deeper investigation.

Abnormal expression of FXR in the kidney and activation of TGF β1, a downstream molecule, is one possible mechanism (Magdy et al., 2018). The Smad signaling pathway induces ECM accumulation, while at the same time, mitochondrial homeostasis also affects cellular aerobic respiration of the carboxylic acid cycle and oxidative phosphorylation, leading to a reduction in ATP production. Abnormal oxidative stress resulting from this process has been shown to induce apoptosis and injury, ultimately leading to glomerulosclerosis. This has been implicated in the pathogenesis and progression of DKD. Not only do these studies provide further experimental evidence to elucidate renal fibrosis in DN, but also provide novel insights into the clinical management of DN.

Materials and Methods

Animal model

The Nantong University’s Animal Care and Use Review Committee (P20230315-004) reviewed and approved each experiment. We purchased db/db mice and FXR knockout mice from the Jackson Laboratory. The four experimental animals were crossed with one FXR+/−db/m male animal and two FXR+/−db/m female: FXR+/+db/db, FXR−/−db/db, FXR+/+db/m, FXR−/−db/m (n ≥3 sample size calculation software). Experiments were performed with FXR−/−db/db males and the other three genotypes from the same cage, 6 months of age. Animals were maintained on normal mouse chow at the Animal Facility of Peking University Health Science Center, animals were kept on normal mouse chow and housed on a 12-hour light/dark cycle under controlled temperature (22−24 °C) and humidity (50–65%) conditions. All mice were euthanized by CO2 inhalation. Weight, blood glucose, and body fat of each group of mice were monitored and recorded monthly.

The collection and analysis of urine’s osmolality

To collect 24-hour urine samples, mice were kept in separate metabolic cages (Tecniplast) with free access to food and water. Body weight, urine output, and water intake were assessed. Prior to assessment, mice were keep in separate metabolic cages for 24 h. The supernatants from the centrifuged urine samples were used to determine the osmolality of the samples using a Micro-Osmometer 3,300 freezing point depression osmometer at 3,000 xg, 4 °C, for 5 min.

ITT and OGTT test

The purpose of the ITT test was to determine insulin sensitivity in mice. Mice were water deprived and weighed at the start of food deprivation. Fasting blood glucose was measured after 4 h of fasting, i.e., 0 min. Intraperitoneal injection of insulin (1 µl/kg dose, 1/3 µl/mL stock solution) was then performed. Blood glucose measurements were taken at 15 min, 30 min, 60 min, 90 min and 120 min intervals. Glucose was administered immediately if any mouse exhibited hypoglycemia or even life-threatening symptoms during the test. The OGTT was performed to assess glucose tolerance in mice. Mice were water deprived and weighed at the start of food deprivation. The sugar solution (5 g/5 mL, one mL of water was added first and then the volume was adjusted) was prepared at the beginning of the experiment. Fasting blood glucose was measured after 4 h of fasting, i.e., 0 min. Glucose was then administered (3 µl/g). Blood glucose measurements were taken at 15 min, 30 min, 60 min, 90 min and 120 min intervals.

Biochemical analyses

The next day, the mice were fasted overnight. Blood was collected from the animal’s tail vein of the animals to measure blood glucose levels using a Freestyle brand glucometer (Roche). The proteinuria, uric acid, urea nitrogen and creatinine levels of the mice were evaluated at Peking University Third Hospital.

In vitro cell culture of primary mesangial cells (P-MC)

Wild-type and FXR knockout out mouse P-MC were cultured in culture with minor modifications. Briefly, male mice (4–6 weeks old) were dissected from the kidney cortex, rinsed, and immersed in an enzyme solution (collagenase: one mL/kidney) for 40 min with continuous agitation at 37 °C. The cell suspension was filtered through a 70 µM to collect the supernatant, which was collected on a 40 µM sieve, and the P-MC cells were grown in DMEM-HG. After 2 to 3 days in culture, the cells hovered around the glomerulus and exchanged fluid for the first time. After approximately one week, the passaged cells develop into a long fusiform shape. Endothelial cells and epithelial cells progressively die out after abour 2 weeks.

Oval mesangial cells emerge. After about 4 weeks, the cells are passed on to the 5th or 6th generation. The membrane cells were purified and tested with cells from the 8th generation to the 12th generation of cells. The FXR agonist CDCA (Sigma Aldrich, St. Louis, MO, USA) was used in initial dose–response tests to determine the does at which CDCA was most effective in promoting FXR protein production while causing the least amount of cytotoxicity. There was no cell damage when P-MC cells were exposed to 100 nM of CDCA. The following experimental conditions were applied to the cells for 24 h prior to RNA extraction and for 48 h prior to protein extraction when both wild-type and FXR−/− mouse P-MC cells were 60–70% confluent: (1) DMSO (vehicle control); (2) five ng/mL TGF; (3) five ng/mL TGF with 100 nM CDCA. The adenovirus α2 titers of FXR (Ad-VP16, University of California, Los Angeles) can be accurately calculated using the KARBER statistical procedure. Cells were subjected to the following experimental conditions for 24 h prior to RNA extraction and for 48 h prior to protein extraction when wild-type and FXR−/− mice P-MC cells were 60–70% confluent: GFP was used as the vehicle control, followed by Ad-FXR2, five ng/mL TGF-1+ GFP, and five ng/mL TGF-1+ Ad-FXR2.

mRNA Isolation and RT-PCR

A commercially available mRNA isolation kit (Biotek, Winooski, VT, USA) was used to isolate total mRNA. Fermentas RevertAidTM First Strand cDNA Synthesis Kit was used to reverse transcribe mRNA samples into cDNA. SYBR Green 1 (Bio-Rad, Hercules, CA, USA) was used in the PCR reaction with cDNA as the template. The SYBR® Premix Ex Taq® II kit (Takara Bio, Inc., Kusatsu, Japan) and the Applied Biosystems 7500 Real-Time PCR System (CXF96; Applied Biosystems, Waltham, MA, USA) were used according to the manufacturer’s protocols. The following primers were used: TGFβ1 forward: 5′-AGCCCGAAGCGGACTACTAT-3′; TGFβ1 reverse: 5′- CTG TGTGAGATGTCTTTGGTTTTC-3′; FXR α forward: 5′- CCG AGAGAAGAACCG AGTT-3′; FXR α reverse: 5′-TAG ATG CCA GGA GAA TAC CAG-3′; FXR β forward: 5′-ATGCAGTTTCAGGGCTTAGAA-3′; FXR β reverse: 5′-CGGGACATT GTTGTATGGG-3′; β-actin forward: 5′-TGTTACCAACTGGGACGACA-3′; β-actin reverse: 5′-GGG GTGTTGAAGGTCTCAAA-3′; 18S forward: 5′-GAA ACGGCTACCACATCCAAG G-3′; 18S reverse: 5′-GCCCTCCAATGGATCCTCGTTA-3′. The PCR reaction was run at 94 °C for 5 min, followed by 35 cycles of 94 °C for 30 s, 59 °C for 30 s, and 72 °C for 30 s, with a final extension of 5 min at 72 °C. RNA levels were normalized to the internal controls 18S and β-actin, and calculated using the 2−ΔΔCt method.

Protein extraction and western blot analysis

Kidney or cellular proteins were solubilized in lysis buffer containing phosphatase inhibitor (P1260, PPLYGEN) supplemented with 0.2 mg/mL of PMSF. Nuclear extracts were extracted with the NE-PER kit (78833; Pierce Biotechnology, Inc., Waltham, MA, USA) according to the manufacturer’s instructions. The bicinchoninic acid assay (P001; Vigorous Biotechnology, Beijing, China) test was used to calculate protein concentrations. A nitrocellulose membrane was then prepared by transferring a total of 20 g of protein, which was combined with loading buffer and separated on a 10% SDS/PAGE gel. After blocking with 5% skimmed milk, the membrane was treated with a variety of primary antibodies for an extended period of time at 4 °C. Mouse anti-FXR (1:1000), rabbit anti-TGF-1 (1:1000), mouse anti-actin (1:1000), rabbit anti-p-Smad3, Smad3, Smad7, and collagen-1, connective tissue growth factor (CTGF) (1:1000) were the primary antibodies used in this work. The membrane was then cleaned and treated with HRP-conjugated secondary antibodies for 1 h at room temperature (Santa Cruz Biotechnology, Dallas, TX, USA). After another wash, the membrane was exposed to Kodak XBT-1 X-ray film to detect antibody binding using a chemiluminescent substrate (sc-2048; Santa Cruz Biotechnology).

Immunohistochemistry (IHC)

For immunohistochemical studies, kidneys were fixed with 4% (wt/vol) PFA in PBS, dehydrated, and embedded in paraffin and subjected to IHC. Kidneys were then sectioned (4 µm) and incubated in the presence of selected primary antibodies overnight at 4 °C. The sections were then treated with α-SMA (1:500), TGFβ1 (1:500), and collagen I (1:500) antibodies (Zhongshan Golden Bridge) coupled to a secondary horseradish peroxidase enzyme for 30 min at 37 °C. Hematoxylin was used to counterstain on the slides.

Immunofluorescence (IF)

Kidney cryosections were mounted on polylysine-coated slides after immersion and fixed in ice-cold acetone for 10 min, washed three times in PBS, permeabilized in 0.1% Triton X-100 in PBS for 10 min, blocked in 0.5% BSA, and incubated with primary antibodies at 4 °C overnight. After washing, sections were incubated for 1 h at room temperature with the appropriate DyLight 488 (green) or DyLight 594 (red)-conjugated secondary antibodies from Jackson ImmunoResearch Laboratories. DAPI was used to stain the nuclei, and images were captured using a confocal microscope. Immunofluorescence labeling was performed on P-MCs in culture after they had been fixed with 4% paraformaldehyde (wt/vol) (PFA) for 15 min on a rocking platform at room temperature.

Chemicals and reagents

TGFβ1 and chenodeoxycholic acid (CDCA) were purchased from Sigma and were used in the study. Anti-farnesoid X receptor (FXR) in mice was purchased from Perseus Proteomics. Primary antibodies against to p-Smad3, Smad3 that were used in the current study were purchased from ABclonal. Anti-Smad7 and anti-collagen I antibodies were purchased from ABclonal. Anti-SHP, CTGF, α-SMA, and anti- succinate dehydrogenase (SDHA) antibodies were purchased from Abcam. Peter Edwards, at the University of California, Los Angeles, kindly provided the data for the adenoviruses expressing FXR α1 (Ad-FXR α1), FXR α2 (Ad- FXR α2), and their control adenoviruses (Ad-VP16).

Statistical analysis

GraphPad program’s Prism program was used for the data analysis. Data are presented as mean ±SEM. A two-tailed Student’s t-test was used to evaluate individual differences. The threshold for statistical significance was P < 0.05.

Results

Loss of FXR reduces obesity and accelerates glucose dysregulation in db/db mice

To investigate whether FXR is involved in the process of renal fibrosis in DN, and to observe the effects of FXR knockout on renal fibrosis in db/db mice, we bred FXR & leptin receptor double knockout mice. Using weight tracking, we observed that body weight was reduced in the FXR−/−db/db mice compared to the FXR+/+db/db mice (Fig. 1A). A significant change was observed in both FXR+/+db/db and FXR+/+db/m mice, which was similar to that seen in previous studies (Fig. 1B). We followed the random glucose monitoring in the double knockout mice and found that there was a significant increase in random glucose in the FXR−/−db/db mice as compared to the FXR+/+db/db mice respectively (Fig. 1B). As shown in Fig. 1C, fat, lean, and water mass were reduced in FXR−/−db/db mice. In contrast, lean and water mass were increased in FXR+/+db/db mice compared to FXR−/−db/db mice (Fig. 1C). Via ITT and OGTT assay revealed that FXR−/−db/db mice had decreased insulin sensitivity compared to FXR+/+db/db mice respectively (Fig. 1D). Diabetes is often caused by hyperglycemia, with urinary glucose exceeding the renal glucose threshold, resulting in a diuretic phenotype through osmotic diuresis. In our study, FXR−/−db/db mice had the highest urine output with the lowest urine osmolality respectively (Fig. 1E). Taken together, loss of FXR in db/db mice reduces obesity and leads to glucose dysregulation.

Figure 1 Effect of FXR knockout on body weight, body fat, blood glucose, insulin sensitivity, urine output and urine osmolarity in db/db mice.

(A) Body weight of four types of mice: FXR+/+db/db (n = 4), FXR−/−db/db (n = 4), FXR+/+db/m (n = 6), and FXR−/−db/m (n = 6). (B) Random blood glucose results. (C) Results of body fat, body fat percentage. (D) Results of ITT and OGTT tests (E) Results of 24 h urine volume and urine osmolality. Data are presented as mean ±SEM.∗P < 0.05,∗∗P < 0.01,∗∗∗P < 0.001 vs. FXR+/+;#P < 0.05,##P < 0.01, ###P < 0.001 vs. db/m.

Loss of FXR accelerates renal injury in db/db mice

The increased levels of urea, creatinine (Cr), uric acid (UA), and proteinuria in the FXR+/+db/db and FXR−/− db/m mice compared to the FXR+/+db/m mice, indicate more severe kidney damage. These blood biochemical markers were significantly increased in the FXR−/−db/db mice compared to the FXR+/+db/db mice (Table 1), indicating more severe renal disease. Immunohistochemistry for α-SMA, TGFβ1, collagen I, and Masson’s staining showed that the renal fibrosis was most severe in the FXR−/−db/db mice (Fig. 2).

Table 1 Biochemical analyses of the mice in the four type mice.

	FXR+/+db/db	FXR−/−db/db	FXR+/+db/m	FXR−/−db/m	
Urea (mmol/L)	10.4 ±1.2a	15.3 ±0.2b	3.5 ±0.4	6.4 ±0.5a	
Cr (umol/L)	24.7 ±4.0a	37.1 ±4.0b	12.8 ±1.3	18.5 ±2.8a	
UA (umol/L)	295.7 ±67.3a	384.0 ±71.2b	160.1 ±12.0	196.2 ±23.0a	
Proteinuria (mg)	2,010.2 ±191.1a	3,412.0 ±162.9b	625.2 ±21.7	1,437.9 ±23.7a	
Notes.

Abbreviations: data are expressed as the mean ±SEM.

a P < 0.01 vs. FXR+/+db/m.

b P < 0.05 vs. FXR+/+db/db,

n, 4-6 per group.

Figure 2 Immunohistochemistry of α-SMA, TGF β1, collagen I and Masson staining shows that loss of FXR accelerates renal injury in db/db mice (n = 3). Magnification: (x400).

Effect of FXR on the expression of renal fibrosis in db/db mice

The pathological mechanisms of renal fibrosis are complex and involve cells, cytokines, extracellular mechanisms, growth factors, and the interplay between these factors. However, most researchers believe that activation of ECM-producing cells and accumulation of ECM are key to the development of renal fibrosis. Collagen I and CTGF have been shown to be aggregation factors. Transforming growth factor 1 (TGFβ1), a cytokine with multiple biological activities known to be a fibrogenic factor that leading to ECM deposition and an important regulator of glomerular sclerosis. A key mediator in promoting ECM production while inhibiting ECM degradation is TGFβ1, which has been shown to play an important role in fibrosis. Consisting of eight different proteins, the Smad family acts as key regulators in the function of multiple signaling pathways via reversible phosphorylation, with excitatory and inhibitory effects respectively. Smad3 and Smad7 are an excitatory–inhibitory cytokine pair. After demonstrating that knockdown of FXR affects kidney fibrosis in db/db mice, kidneys were harvested from four types of mice were extracted and the expression of the related fibrosis factors (collagen I, collagen IV, CTGF and TGFβ1-Smad signaling pathway (Smad3, Smad7) and the FXR chaperone factor small heterodimer partner (SHP) were examined. We found that the expression and protein levels of the related fibrosis factor, collagen I, TGFβ1 and Smad3 were significantly increased in FXR−/−db/db mice compared with those in FXR+/+db/db mice, whereas the levels of SHP and Smad7 were significantly decreased (Fig. 3A, Fig. 3B). In conclusion, FXR may be involved in the development of renal fibrosis in DN through the regulation of the TGFβ1-Smad signaling pathway in vitro.

Figure 3 Real-time PCR and Western blot results of associated factors in four types of mice.

(A) Changes in the mRNA levels of fibrosis factor, TGF β1-Smad pathway-related factors, and FXR chaperone SHP in four types of mice (n = 3). (B) Changes in protein levels of fibrosis factor, TGF β1-Smad pathway-related factors and FXR chaperone SHP (n = 3). Data are presented as mean ±SEM. ∗P < 0.05,∗∗P < 0.01,∗∗∗P < 0.001 vs. FXR+/+;#P < 0.05,##P < 0.01 vs. db/m.

FXR represses fibrosis in primary mesangial cells via downregulation of Smad3 and upregulation of Smad7 expression

To study FXR regulation of TGFβ1 in vitro, we first cultured and identified murine P-MC. In response to TGF β1 stimulation of P-MCs in FXR+/+ mice, mRNA and protein levels of fibrogenic factor, TGFβ1, and Smad3 were significantly increased and Smad7 levels were decreased. These effects were reversed by CDCA. Moreover, CDCA further enhances the expression and protein levels of FXP and SHP, which were already increased by TGFβ1 (Fig. 4A). As for the P-MCs of FXR−/− mice, TGFβ1 resulted in up-regulation of collagen I and TGFβ1 expression and protein levels, which was unaffected by the addition of CDCA (Fig. 4B). In addition, adv-FXR α2 was found to down-regulate related fibrosis factors, TGFβ1 and Smad3, and up-regulate Smad7 with or without TGFβ1-induced in P-MCs cells (Fig. 4C, Fig. 4D). Taken together, these data indicate that FXR-mediated Smad3 inhibition and Smad7 promotion are FXR-specific, and FXR suppresses TGFβ1-induced cellular fibrosis via downregulation of Smad3 expression and upregulation of Smad7 expression.

Figure 4 FXR suppresses fibrosis in P-MC via downregulation of Smad3 and upregulation of Smad7 expression.

(A, B) After treatment with TGF β1 (five ng/mL) for 24 h in FXR+/+ mice (A) and FXR−/− mice (B), P-MCs were treated with CDCA (100 nM) or DMSO for another 24 h. Then, the mRNA expression of collagen I, CTGF, TGF β1, FXR, SHP, Smad3, and Smad7 were determined by qRT-PCR (n = 3), and the protein expression of fibrosis factors, FXR, SHP, p-Smad3/Smad3, and Smad7 were determined by Western blot (n = 3).∗P < 0.05,∗∗P < 0.01, ∗∗∗P < 0.001 vs. TGF β1-CDCA; #P < 0.05,##P < 0.01,###P < 0.001 vs. TGF β1+CDCA. (C, D) After treatment with adv-FXR α2 (10 MOI) or GFP for 24 h in FXR+/+ mice, P-MCs were treated with TGF β1 (five ng/mL) for another 24 h. Then, the mRNA expression of collagen I, CTGF, TGF β1, Smad3, and Smad7 were determined by qRT-PCR (n = 3) (C), and the protein expression of fibrosis factors, p-Smad3/Smad3, and Smad7 (D) were examined by Western blot (n = 3).∗P < 0.05, ∗∗∗P < 0.01, ∗∗∗P < 0.001 vs. Ctrl adv-GFP; #P < 0.05,##P < 0.01,###P < 0.001 vs. adv-GFP.

Loss of FXR in db/db mice is associated with increased energy expenditure (EE)

Because FXR−/− mice have lower body mass index but similar food consumption, it is possible that the absence of FXR leads to increased EE. Therefore, we used indirect calorimetry to determine oxygen uptake and CO2 production. Compared with FXR+/+db/db mice, FXR−/−db/db mice had increased VO2 (Fig. 5A), VCO2s (Fig. 5B), and EE (Fig. 5C). In addition, the respiratory quotient (RQ) remained unchanged in the range of 0.74−0.79 (Fig. 5D), while the activity level increased in FXR−/−db/db mice (Fig. 5E), indicating that the primary fuel source of the mice was fatty acids. In the tricarboxylic acid cycle of aerobic respiration, SDHA, a marker enzyme for mitochondria, was significantly decreased in FXR−/−db/db mice (Figs. 5F–5G). Taken together, these results suggest that FXR−/−db/db mice have increased EE, which may at least partially account for the reduced adiposity in these mice.

Figure 5 FXR deficiency increases energy expenditure.

Four types of mice were fed for 6 months (n = 3 per group), oxygen (O2) consumption (A), carbon dioxide (CO2) production (B), energy exchange rate (C), RQ (D), and activity (E) were determined. ∗P < 0.05,∗∗P < 0.01. (F) Immunohistochemical detection of SDHA in mice with four genotypes. (G) SDHA protein levels were determined in four types of mice.

Discussion

Renal fibrosis stands as a hallmark of many progressive kidney disorders, serving as a pivotal determinant of renal failure and a reliable prognostic indicator (Li, Fu & Liu, 2022). Within this intricate mechanism, the dominant role of TGF in fibrogenesis is undeniable. Among the three TGF isoforms found in mammals, TGF1 stands predominant. Our findings, in alignment with prior research, indicate the pivotal role of TGFβ1 in renal fibrosis onset (Chen et al., 2021; Gifford et al., 2021). This signaling promotes ECM production through type I and type II serine/threonine kinase receptors, with the renowned Smad signaling pathway acting as a central player in kidney fibrosis progression (Zeitlmayr et al., 2022). Key to fibrogenesis is Smad3’s heightened activity, which in conjunction with Smad7 downregulation, triggers myofibroblast activation, ECM accumulation, and its reduced breakdown. Therapeutically, Smad7 overexpression has shown promise in renal fibrosis models. Counteracting Smad3, Smad2 serves as a protective entity within the TGF/Smad signaling framework.

Meanwhile, FXR, a bile acid receptor primarily expressed in the liver and intestine, plays a decisive role in bile acid production and circulation while also acting as a metabolic nuclear receptor (Chiang & Ferrell, 2020). In the renal domain, studies spotlight FXR’s potential as a therapeutic asset for DN (Guo, Xie & Zhang, 2023; Luan et al., 2022), with our findings indicating its antifibrotic effect via Smad3 downregulation and Smad7 upregulation. FXR’s pivotal role in DN pathogenesis, especially in the db/db mouse model, is underscored by our findings. Notably, FXR’s activation might counteract fibrosis in primary mesangial cells influenced by exogenous TGFβ1, restoring the balance of the TGFβ1-Smad signaling pathway. While FXR activation boasts multiple benefits, its potential weight-regulating effects, especially in hemodialysis patients, warrant careful consideration given the link between weight gain and survival in the initial year of dialysis therapy (Xu et al., 2022). Consequently, strategies curtailing weight gain should proceed with caution, ensuring no unintended reductions in dietary calorie and protein intake. The implications for hemodialysis patients are clear: weight loss isn’t always beneficial.

Aerobic respiration, a quintessential cellular process, chiefly occurs within the mitochondria, the cellular powerhouse. Here, organic molecules like glucose are methodically oxidized in an oxygen-rich environment to yield CO2, H2O, and, most vitally, a bounty of ATP—the cell’s primary energy currency. The tricarboxylic acid cycle, operating in the mitochondrial matrix, coupled with oxidative phosphorylation, forms the crux of eukaryotic oxidative metabolism, churning out indispensable energy molecules like NADH and FADH2. Intriguingly, our research draws a compelling link between FXR knockout and noticeable shifts in the metabolic metrics, encompassing oxygen consumption, carbon dioxide generation, and energy conversion rates in db/db mice. FXR, as a sentinel in energy metabolism orchestration, could modulate pathways influencing energy metabolism. Disturbances in kidney morphology and function invariably herald cellular energy metabolic anomalies, primarily manifesting as mitochondrial metabolism aberrations (Weaver, Healy & Wingert, 2022). Our observations revealed augmented activity in FXR−/−& db/db mice compared to their FXR +/+& db/db counterparts, leading to heightened oxygen utilization and energy conversion. This insinuates FXR’s pivotal role in modulating aerobic respiration in the db/db mouse model.

Zooming in on the tricarboxylic acid cycle, SDHA emerges as an exceptional entity. As the lone membrane-integrated multisubunit enzyme in aerobic respiration, it bridges the gap between oxidative phosphorylation and electron transport. SDHA channels electrons to the respiratory chain, playing an instrumental role in mitochondrial efficiency (Fujiwara et al., 2021). Our immunohistochemical analyses starkly highlighted diminished SDHA levels in FXR−/−& db/db mice compared to FXR+/+& db/db mice, reinforcing the hypothesis that FXR knockout might be disrupting mitochondrial balance through its influence on aerobic respiration, particularly in the context of DN. Recent explorations, centered on mice on a high-fat diet post-nephrectomy, revealed that FXR agonists could rejuvenate the mitochondrial respiratory chain (Bhatia et al., 2022). Given mitochondria’s pivotal role in respiratory metabolism, our immunohistochemical examinations further cement the prospective interplay between FXR and the regulation of the respiratory metabolic chain. Delving deeper into the mitochondrial processes, FFA metabolism, or β-oxidation, primarily unfolds within these organelles. FXR activation seemingly curtails fatty acid metabolic β-oxidation, thereby rejuvenating the respiratory chain. These insights echo the patterns we discerned in our research. Yet, the underlying mechanics of how the absence of FXR genes shapes the regulation of mitochondrial balance via aerobic respiration remains enigmatic. A promising avenue for further exploration could lie in modulating fatty acid β-oxidation.

Our study illuminates the pivotal role of FXR in DN, ushering in a new avenue for therapeutic interventions. The potential for harnessing FXR agonists may pave the way for innovative treatments that mimic the natural activity of FXR, offering an avenue for alleviating symptoms in patients with DN. Additionally, by focusing on enhancing mitochondrial function, considering personalized medicine based on genetic markers, and integrating FXR modulation with current treatments, a comprehensive approach to disease management emerges.

However, the study has some limitations. First, the in vivo conditions might not encompass the entire spectrum of human DN. Secondly, FXR may affect other intercellular signaling, which was not further analyzed in this study. However, the findings present a promising foundation for further research and underline the potential of FXR as a therapeutic target. The consistency and robustness of our data point towards its broad applicability, suggesting a strong generalizability and potential for significant clinical impact.

Conclusions

In conclusion, this experiment demonstrates that FXR plays an important role in the pathogenesis of DN in the db/db mouse. In db/db mice, deletion of the FXR gene results in decreased insulin sensitivity, a polyuric phenotype with more severe fibrosis. Activation of FXR in primary mesangial cells can ameliorate fibrosis induced by exogenous TGFβ1. FXR may negatively regulate TGFβ1 expression and restore the balance of the TGFβ1-Smad signaling pathway to ameliorate renal fibrosis. In the present study, we also found that deletion of the FXR gene resulted in decreased expression of mitochondrial SDHA by aerobic respiration in db/db mice, suggesting that FXR may be involved in the regulation of DN by aerobic respiration. The development of this project will help to further elucidate the mechanism of FXR involved in DN and provide a theoretical basis and potential targets for the prevention and treatment of clinical DN and the development of new drugs.

Supplemental Information

Supplemental Information 1 ARRIVE Checklist

Click here for additional data file.

Supplemental Information 2 Supplementary Figures

Click here for additional data file.

Additional Information and Declarations

Competing Interests

Author Contributions

Animal Ethics

Data Availability

The authors declare there are no competing interests.

Yuxiang Qiu conceived and designed the experiments, performed the experiments, analyzed the data, prepared figures and/or tables, authored or reviewed drafts of the article, and approved the final draft.

Ningsu Kang performed the experiments, prepared figures and/or tables, and approved the final draft.

Xi Wang performed the experiments, prepared figures and/or tables, and approved the final draft.

Yao Yao performed the experiments, authored or reviewed drafts of the article, and approved the final draft.

Jun Cui performed the experiments, authored or reviewed drafts of the article, and approved the final draft.

Xiaoyan Zhang conceived and designed the experiments, analyzed the data, authored or reviewed drafts of the article, and approved the final draft.

Lu Zheng conceived and designed the experiments, analyzed the data, prepared figures and/or tables, and approved the final draft.

The following information was supplied relating to ethical approvals (i.e., approving body and any reference numbers):

The Nantong University’s animal care and use review committee approved the study.

The following information was supplied regarding data availability:

The raw data is available at Figshare: Zhen, Lu (2023). Original data.rar. figshare. Journal contribution. https://doi.org/10.6084/m9.figshare.22698508.v2.

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
