# Peer review of "Loss of Farnesoid X receptor (FXR) accelerates dysregulated glucose and renal injury in db/db mice"

_PeerJ, doi:10.7717/peerj.16155_

## Round 0.1 · original submission · Minor Revisions

This study utilizes FXR knockout mice and investigates the effects of FXR deficiency on body weight, glucose regulation, renal function, and energy expenditure. Overall, the study provides valuable insights into the role of FXR in diabetic nephropathy. However, several issues need to be addressed before this manuscript can be considered for publication.

Please respond and make appropriate revisions based on the reviewers' suggestions and my comments (below). This will greatly improve the quality of this manuscript.

My comments:
1. Abstract: [FXR treatment with adenovirus produced similar results]: The meaning of this sentence is unclear and needs to be rewritten so that readers can comprehend it.
2. [The possible mechanism is abnormal … aerobic respiration]: should be revised to [The possible mechanism is that abnormal expression of FXR activates TGFβ1 and then induces ECM accumulation via the classical Smad signaling pathway, whereas FXR affects mitochondrial function via steady-state mitochondrial function].
3. Figure 1A was not clearly explained in the Results section.
4. Line 315: Figure 4C and 4B should be Figure 4A and 4B.
4. Line 317: Figure 3C and 3D should be Figure 4C and 4D. Please confirm this.
5. The English of this article should be edited and improved. The text contains grammatical errors, awkward sentence structures, and unclear phrasing. Improving the language will enhance the readability and overall quality of the manuscript.
5.1. For example, a increase in blood glucose or an increase in blood glucose?
5.2. [For the purpose of collecting] or [to collect]?
5.3. grown in in culture?
5.4. Synthess or Synthesis?
5.5. Cellural or cellular?
5.6. [to observe the effects] or [and to observe the effects]?
5.7. [The common conclusion … a trustworthy predictor of prognosis] should be revised to [The common conclusion for almost all progressive kidney disorders is renal fibrosis. Additionally, it is a crucial factor in determining renal failure and a trustworthy predictor of prognosis]?

**Language Note:** The Academic Editor has identified that the English language must be improved. PeerJ can provide language editing services - please contact us at [email protected] for pricing (be sure to provide your manuscript number and title). Alternatively, you should make your own arrangements to improve the language quality and provide details in your response letter. – PeerJ Staff

Reviewer 1 ·

Basic reporting

1.This study investigates the crucial role of Farnesoid X receptor (FXR) in the context of diabetic kidney disease (DKD), which is a significant cause of end-stage renal disease. This contributes to the field by examining a potentially key biological process related to a major health issue. A novel and comprehensive approach is taken in the research by using FXR & leptin receptor double knockout mice, monitoring various indicators over a period of 6 months. This long-term, multifaceted approach can yield insights not only about immediate effects but also about the progression over time. The findings provide evidence of the critical role FXR plays in mesangial cell regulation in diabetic nephropathy, elucidating a mechanism involving the TGFβ1 pathway and Smad signaling pathways. These mechanistic insights are significant for understanding the disease at a molecular level. The study also touches upon the impacts of FXR on metabolic functions, like aerobic respiration and energy conversion, suggesting an overarching role for FXR in maintaining metabolic homeostasis and kidney function. However, there are areas where the manuscript could be improved for clarity and coherence.Some basic spelling and language improvements are needed. For example:
Line 31 “Samd3”
Line 37 “activity”,
Line 41-43 the sentence is not clearly expressed
Line 280 “which means indicating”
……

2. The citations in the references are very old, none of which is even within the last 5 years, which seriously affects the research value of the manuscript, and a large number of references need to be supplemented and replaced, especially in the background and discussion section.

3.The resolution of the pictures needs to be improved, and it is necessary to add scale marks to the pathological Figures.

Experimental design

4.It would be beneficial to clearly describe the methods, variables observed, and more specific results. Addressing these issues could improve the readability of the manuscript and make the findings more accessible to a broader audience.
5.The description of "mRNA Isolation, RT-PCR, and Real-Time PCR" in the Methods section has sufficient details and information for replication.
6.There seems to be a repetition of procedures in the "Respiration Collection and MRI Analysis" section. Make sure to edit one of them out if it's not necessary or clearly distinguish between them if they are different.

Validity of the findings

7."And fibrosis was to be understood by Masson stain, assessing the situation of the resuscitation case by aerobic respiration by SDHA stain immunohistochemistry." - This sentence is a little unclear. It seems to be saying that the study aimed to understand fibrosis through the use of Masson's stain and to assess "the situation of the resuscitation case" (an unclear phrase) through SDHA stain immunohistochemistry, which assesses aerobic respiration.
8."P-MC from FXR -/- mice treated with the FXR agonist CDCA showed no significant differences in related indices." - This sentence is a bit unclear. It could be more specific about what "related indices" refers to.
9."lean and water mass in FXR+/+db/db mice was increased compared to FXR+/+db/db mice respectively" seems incorrect as it compares the same group of mice with itself.
10.You should replace "The data are shown as mean SEM" with "The data are shown as mean ± SEM" to be more clear.
11.Conclusions should be properly stated and should relate to the original question of the investigation. The manuscript begins the discussion section with a large paragraph of background and does not make a correlation with the expressed meaning of the results.

Reviewer 2 ·

Basic reporting

1.This stuy confirms the significant role of FXR in the development of diabetic nephropathy in db/db mice. FXR gene deletion results in decreased insulin sensitivity and more severe fibrosis, accompanied by polyuria. Activation of FXR in mesangial cells can improve fibrosis induced by TGFβ1. FXR may negatively regulate TGFβ1 expression and restore balance to the TGFβ1-Smad signaling pathway, thereby ameliorating renal fibrosis. Additionally, the study finds that FXR gene deletion leads to decreased expression of SDHA, indicating a role of FXR in regulating diabetic nephropathy through aerobic respiration. These findings contribute to a better understanding of the mechanism by which FXR is involved in diabetic nephropathy and offer potential targets for prevention, treatment, and the development of new drugs in clinical diabetic nephropathy. Although the manuscript is generally understandable, there are many poor spelling and grammar mistakes in the article. Please find a professional polishing agency to polish this article.
2.The last sentence in the paragraph of Introduction starting with "Farnesoid X receptor (FXR)..." is very long and difficult to follow, leading to a logical disconnect.
3.Abbreviations that appear for the first time in the text need to indicate the full name, such as db/db mice and ECM accumulation.
4.It is necessary to update the references, especially in the past three years.

Experimental design

5.It has been reported that FXR activation can protect cells from ROS-induced oxidative stress caused by FFA free fatty acids and endoplasmic reticulum stress. FXR activation can reduce ROS and ameliorate FFA-induced kidney injury, while FXR agonist CA may induce recovery of mitochondrial respiration Chain reactions [12]. The author needs to point out the differences between this study and previous studies.
6.The relationship between some concepts is not fully explained or is unclear. For instance, the role of FXR in DKD is not well explained in relation to the previously discussed pathophysiology of the disease. This can be improved by giving a more direct link or explanation.
7. The phrase "because it can be superphysiological level of farnesoic activation named farnesene derivative" is not clear and seems to lack important context. It may need additional information or rephrasing.
8.The authors should provide statistical analysis for all experiments, including sample sizes, replicates, and details on statistical tests used. It is essential to include these details to ensure the robustness of the results.

Validity of the findings

9.The author concluded that “Loss of FXR in db / db mice increases energy expenditure”. However, it does not follow that FXR affects mitochondrial function through steady-state aerobic respiration. Please modify the description of these results.
10.Although the discussion takes up a lot of space in this article, the discussion of this study is not well centered around the findings of the study, especially the function of FXR on mitochondrial function.
11.The paper would benefit from a more detailed discussion section about how the finding could be translated into therapeutic strategies.
12.The resolution of the pictures in the article needs to be improved, and ICONS need to be added to the immunohistochemical and immunofluorescence diagrams in the article.

Additional comments

13.limitations of the study regarding in vivo studies and the generalizability of the findings.

---

## Round 0.2 · Minor Revisions

Even though the Reviewers' concerns were addressed, additional efforts were still needed to further improve the quality of this paper. In particular, the writing about Figure 5 needs to be fleshed out and improved as follows:

1. Lines 332-324: [In FXR-/-db/db mice, VO2 (Fig. 5A), VCO2s (Fig. 5B), as well as EF (Fig. 5C), unchanged respiratory quotient (RQ) (Fig. 5D), and increased activity (Fig. 5E) compared to FXR+/+db/db mice]:
The meaning of this sentence was quite unclear and it failed to make explicit how VO2, VCO2s, and EF, as well as RQ, are different between the two types of mice. The results indicated that the rate of oxygen consumption, the rate of carbon dioxide production, and the rate of energy conversion were increased in FXR-/-& db/db mice (Fig. 5A, 5B and 5C). However, please carefully clarify this and make appropriate modifications.
2. [increased activity] means what? Please explain. Also, the results shown in Figures 5D and 5E should be described in a separate sentence. 

3. [RQ was in the range of 0.74-0.79 range]: Please delete the second [range].
4. Line 292: Delete [which].
5. Line 174: Delete [a].
6. Line 130: Change [for] to [into].
7. Line 111: [farnesoic cascade]: This expression is not commonly written. Please carefully clarify and modify this.
8. Line 83: [Smad protein is sole regulatory substrate of TGFβ1]: Change [sole] to [a].
9. Line 30 and 71: Delete [an].
10. Line 84: [is a sign] should be [are signs].

---

## Round 0.3 · Minor Revisions

Unfortunately, my concerns regarding the writing of Figure 5 have still not been adequately addressed. 

In the Results section, the authors stated that, compared with FXR+/+db/db mice, FXR-/-db/db mice had decreased VO2 (Fig. 5A), VCO2s (Fig. 5B), and EE (Fig. 5C). SDHA, a marker enzyme for mitochondria, was significantly elevated in FXR-/-db/db animals (Fig. 5F-G).

However, in the Abstract section, they claimed that:
> Compared with FXR+/+& db/db mice, the rate of oxygen consumption, the rate of carbon dioxide production, and the rate of energy conversion were increased in FXR-/-& db/db mice, whereas the SDHA succinate dehydrogenase, a marker enzyme for aerobic respiration, was significantly decreased.

It seems that the Abstract correctly reflects the results of the study, while the writing of the Results section could be considered an error that occurred in the drafting. These inconsistent descriptions need to be carefully confirmed and revised.

In addition, in Figure 5: The labeling of the groups (5A, B, C, D, and E) does not seem to match the presentation of the results. Please check carefully here and make appropriate revisions.

---

## Round 0.4 · accepted · Accept

Through a series of revisions, this study has been strengthened and enhanced. I evaluated the revised version. My and the reviewer's concerns have been well addressed. I am happy with the current version. I believe that this revised manuscript is ready for publication.